# Genomic Diversity and Phenotypic Variation in Fungal Decomposers Involved in Bioremediation of Persistent Organic Pollutants

**DOI:** 10.3390/jof9040418

**Published:** 2023-03-29

**Authors:** Jiali Yu, Jingru Lai, Brian M. Neal, Bert J. White, Mark T. Banik, Susie Y. Dai

**Affiliations:** 1Systems and Synthetic Biology Innovation Hub, Texas A&M University, College Station, TX 77843, USA; jialiyu@tamu.edu (J.Y.);; 2Department of Plant Pathology and Microbiology, Texas A&M University, College Station, TX 77843, USA; 3USDA Forest Service, Northern Research Station, Center for Forest Mycology Research, Madison, WI 53726, USA

**Keywords:** agaricomycetes, decolorization, bioremediation, lignin-degrading genes, comparative genomics

## Abstract

Fungi work as decomposers to break down organic carbon, deposit recalcitrant carbon, and transform other elements such as nitrogen. The decomposition of biomass is a key function of wood-decaying basidiomycetes and ascomycetes, which have the potential for the bioremediation of hazardous chemicals present in the environment. Due to their adaptation to different environments, fungal strains have a diverse set of phenotypic traits. This study evaluated 320 basidiomycetes isolates across 74 species for their rate and efficiency of degrading organic dye. We found that dye-decolorization capacity varies among and within species. Among the top rapid dye-decolorizing fungi isolates, we further performed genome-wide gene family analysis and investigated the genomic mechanism for their most capable dye-degradation capacity. Class II peroxidase and DyP-type peroxidase were enriched in the fast-decomposer genomes. Gene families including lignin decomposition genes, reduction-oxidation genes, hydrophobin, and secreted peptidases were expanded in the fast-decomposer species. This work provides new insights into persistent organic pollutant removal by fungal isolates at both phenotypic and genotypic levels.

## 1. Introduction

Fungi represent one of the most diverse organismal groups, with a critical role in global carbon cycling. It is estimated that over 2.2 million fungal species exist worldwide, based on recent technologies [1]. The fungal capacities to degrade organic carbons have prompted bioremediation applications, where the decomposition of a broad range of environmental organic compounds has been studied [2]. Fungi play critical roles in the ecosystem, where fungi that adapt to environmental changes and stresses are recognized as functional groups such as stress tolerators and decomposers [3]. Decomposer fungi have been utilized to remediate various environmental contaminants due to their capacity to respond to environmental stresses and produce redox enzymes that can catalyze breaking down covalent chemical bonds. To date, most documented bioremediation fungi are in the phyla of *Ascomycota* and *Basidiomycota*, with some reported in the subphylum *Mucoromycotina* and a few in other phyla [2]. Wood-decaying basidiomycetes are classified into white-rot and brown-rot fungi due to differences in their mechanisms for breaking down wood. Brown-rot fungi use both an oxidative mechanism involving the generation of highly reactive hydroxyl radicals and a hydrolytic mechanism with hydrolases to degrade cellulose and hemicellulose, while white-rot fungi use a combination of oxidative and hydrolytic mechanisms to degrade lignin and other components of wood [4,5]. White-rot fungi are especially adept at deconstructing lignin, a recalcitrant biopolymer that resembles the properties of environmental persistent organic pollutants (POPs) [4]. The known lignin decomposition enzymes and oxidoreductases (i.e., laccases, oxygenases, and peroxidases) have been utilized to detoxify environmental toxins [6]. For instance, the decolorization rate of azo dye was significantly improved with the presence of lignin, which induced lignin decomposition enzyme production [7,8]. *Basidiomycota* and *Ascomycota* are the most prevalent fungal phyla involved in wood decay and thus have been studied mainly for bioremediation capacities [9]. Currently, the most commonly studied strains of wood-decaying basidiomycetes fungi for bioremediation are members of the genera *Phanerochaete*, *Pleurotus*, *Trametes*, and *Gloeophyllum* [10,11,12,13]. 

Many synthetic POPs are widely present in the environment after release. These chemicals can be transported through the atmosphere and water and end up deposited into sediments and soils, which negatively impacts the ecosystem and human health [2]. As fungi respond to environmental stress to survive, they can naturally utilize the POPs as a nutrition source and/or eliminate them as a defense and detoxification mechanism [14]. The low specificity of many fungal enzymes results in a broad bioremediation capacity as the fungi can co-metabolize structurally diverse compounds. One fungus or a consortium can degrade various chemicals such as petroleum hydrocarbons, phenolic compounds, organochlorines, pesticides, synthetic polymers, and others. During the decomposition or detoxification process, enzymes such as laccase, peroxidase, cytochrome P450, and nitroreductases work synergistically to biotransform carbon, nitrogen, phosphorus, and other elements [15]. 

The investigation of specific traits in fungi has been studied in the context of taxonomic resolution, with a focus on genes responsible for the decomposition of organic compounds [3]. For example, the Class II peroxidase gene (POD) family, one of the most important fungal gene families for lignin degradation, is expanded in the ancestors of the class Agaricomycetes but not commonly found in other clades [4]. It is thus imperative to analyze the fungal gene functions at the taxonomic levels of class or smaller [3]. Understanding how fungi adapt to their environment and how this adaptation relates to bioremediation remains challenging. In this study, we investigated the decolorization phenotypes of various Agaricomycetes in combination with genome-wide gene family analysis to understand potential mechanisms for fungal environmental adaptation variations and assess their utility for bioremediation. 

## 2. Materials and Methods

### 2.1. Fungal Isolate Cultivation

Firstly, 320 basidiomycete isolates were selected from the USDA Forest Service, Center for Forest Mycology Research culture collection, located at the Forest Products Laboratory, Madison, Wisconsin, United States. The molecular identification of fungal species for the collected fungal isolates was performed upon designing the study, as described below. Isolates were chosen to cover a wide taxonomic range of wood-decay fungi based on recent phylogenetic rankings [16,17], including both white-rot- and brown-rot-causing fungi. Selected isolates were stored on 2% malt extract agar (MEA) media slants in glass tubes at 4 °C. Fungal mycelia were transferred from the MEA slants onto Petri plates (90 mm × 15 mm) containing potato dextrose agar (PDA). The cultures were then incubated at 30 °C until the mycelia covered the entire surface of the agar and then served as starter plates for use in growth experiments. The collection information for each isolate is listed in Appendix A and visualized by ‘tmap’ package [18]. 

### 2.2. Phylogenetic Analysis

The identity of each isolate selected was verified via the amplification and sequencing of the entire ITS region of the ribosomal DNA gene. Primers ITS-4 [19] and ITS-1F [20] were used for amplification and sequencing following previously established protocols [21]. The procedure was modified for the use of tip swipes of fungal cultures as template DNA by the addition of 3 μL of water to each master mix to replace the liquid present in the purified DNA template that was not used. Following Sanger sequencing, the trimmed sequences were identified via a BLAST search of the NCBI database. Sequences identified as the same taxon were then compared among themselves to confirm conspecificity. The entire ITS regions of the sequences were used to determine the phylogeny distances of the fungal isolates (Appendix A). Specifically, ITS sequences were subjected to multiple sequence alignment (MSA) by Clustal-Omega version 1.2.4 [22]. We then inferred a maximum likelihood phylogenetic tree with the resulting alignments using RAxML 8.2.12 under the ‘GTRGAMMA’ model with 100 bootstrapping [23]. The phylogenetic tree was visualized by the R package ‘ggtree’ [24]. 

### 2.3. Growth Rate and Dye-Decolorization Capacity Assay

For growth rate and dye-decolorization assays, a single 7 mm agar plug was inoculated onto a testing plate with the following method. Agar plugs were cut at a distance of one centimeter (cm) from the growing edge of PDA starter plates using an inverted 1 mL sterile pipet tip. Subsequently, an agar plug with mycelia was aseptically placed in the center of the appropriate test plate and incubated at 30 °C. For growth rate studies, PDA medium was used, and mycelial extension was measured daily from 2 to 10 days after inoculation. To facilitate growth measurements, we generated a scaled empty plate to serve as a circular ruler with four concentric circles marked with permanent ink, at 1 cm, 2 cm, 3 cm, and 4 cm diameters (Figure 1A,B). Fungal growth was rated as zero if no growth was observed, 1–4 depending on the cm of radial growth measured, or 5 if the entire plate was covered. If the mycelia expanded to the midpoint between two circle zones, a score of 0.5 was assigned. Each isolate had two replicated plates, with the measurements from both plates being averaged to represent the growth rate of the isolate.

For the dye-decolorization assay, PDA medium was modified with the addition of 50 mg/L Direct Red 5B (PDA+DR5B). Direct Red 5B (Sigma-Aldrich, St. Louis, MO, USA) is an azo dye that is widely used in the textile industry, leading to water pollution. The zone of decolorization was measured with the same method as described for growth measurements from day 2 to day 10 after inoculation. The capacity to decolorize was rated null if no decolorization occurred, I–IV if the decolorization zone extended for 1–4 cm, and V if the whole plate was decolorized (Figure 1C). The decolorization assay was replicated twice for each isolate.

### 2.4. Dye Degradation Trait Variation Analysis

Following the completion of the decolorization assay, the isolates were divided into two groups based on the rate at which decolorization occurred. The 320 isolates were clustered by hierarchical clustering using the Euclidean distance of the 10-day period of dye-decolorization rates. They were classified into two clusters: a high-decolorization group and a low-decolorization group, with the former exhibiting a faster rate than the latter. In each species, dye-decolorization indexes of all isolates within species were compared to the whole library by the non-parametric Mann–Whitney Wilcoxon test. The statistically significant difference was determined by the p-value cutoff of 0.05. The within-species variance was determined by the standard deviation of the dye-decolorization indexes from all isolates within a species. The correlation coefficient between the growth index and dye-decolorization index in the selected isolates was calculated by Pearson’s correlation coefficient. The dependence of growth and dye-decolorization across the fungal library was determined by the Chi-square test. All statistical tests were performed with the R package ‘stats’ and plotted with ‘ggplot2’ [25]. 

### 2.5. Genome-Wide Gene Family Evolutionary Analysis

Reference genomes from Agaricomycetes species were obtained from Mycocosm [26], with details listed in Appendix A. Among the top-ranked isolates, we identified 11 species that contain high-quality annotated genomes from JGI published sequences. To minimize the analysis variance, we excluded the NCBI unannotated assemblies. The protein sequences from filtered gene models with the longest isoform of each gene were used to identify orthogroups by OrthoFinder version 2.5.4 [27,28]. The species phylogeny was inferred by the genome-wide orthogroups across the eight species with the default settings in OrthoFinder. The rooted species tree was scaled according to the divergence time between *Polyporales* and *Auriculariales* at 249.9497 million years ago, as referred to by Krah et al. (2018) [29]. Functional annotations for the orthogroups were assigned by InterproScan version 5.50 [30] using a representative protein sequence from each orthogroup. Orthogroups with the same function were grouped together as a functional gene family and the gene content in each known functional gene family was summarized. The genome-wide gene family expansion and contraction were analyzed using CAFE 5.0 [31]. A lambda value estimating the evolutionary rate of all species was calculated and the significant expansion or contraction of each orthogroup was determined by a p-value cutoff of 0.05. The phylogenetic tree and gene family evolutionary results were visualized in R packages ‘ggtree’ and ‘ggplot2’ [24,25].

### 2.6. Identification of Gene Families of Genes Encoding Wood-Decay-Related Enzymes

The gene families encoding carbohydrate-active enzymes (CAZymes) and wood-decay redox enzymes were identified using the method from Floudas et al. (2015) [32]. Briefly, orthogroups identified by OrthoFinder were assigned with InterPro or Pfam accessions. Wood-degrading genes with an InterPro or Pfam ID were used to identify the orthogroups with the annotated InterPro or Pfam IDs. For the genes without InterPro IDs, the represented orhtogroup sequences were BLAST searched against the indicated genes with an E-value cutoff of 1 × 10^−30^. The matched orthogroups were assigned to the corresponding gene families. The gene content of each gene family was summarized by clustering the orthogroups with the same functional annotation. Enzyme commission (EC) numbers were assigned to CAZymes using the CAZyme database (http://www.cazy.org/, accessed on 20 March 2023) [33]. The InterPro IDs of significant gene families were BLAST searched against Uniprot with default parameters to identify their EC numbers in the enzyme database [34].

## 3. Results

### 3.1. Mapping of Fungal Isolates

In this study, a total of 320 basidiomycete fungal isolates were collected from Canada, the United States, and China. Among the 320 isolates, 314 were collected throughout the United States, including Alaska and Hawaii (Figure 2A). Three isolates were collected from Ontario, Canada. One isolate was from China and two were from unknown locations. Except for 26 isolates from unknown hosts, 312 isolates were collected from live or dead wood, with the majority being hardwood trees (Appendix A). The taxa of the fungal isolates were identified by internal transcribed spacer (ITS) sequences. The 320 isolates were assigned to 74 species, in which *Wolfiporia cocos*, *Trametes versicolor*, *Cerrena unicolor*, and *Irpex lacteus* were the four biggest groups, including 14, 10, 10, and 10 isolates, respectively. All species were in the class of *Agaricomycetes,* with 46 in Polyporales, 9 in Hymenochaetales, 6 in Agaricales, 4 in Corticiales, 2 in Russulales, and 1 in Atheliales, Auriculariales, Boletales, Cantharellales, Gloephyllales, Phallomycetidae, and Trechisporales (Figure 2B). Their phylogenetic relationship is shown in Figure 2B using the entire ITS region from each species. Of the 320 isolates in our study, 256 were white-rot fungi, 56 were brown-rot fungi, and 8 isolates belonging to the species *Athelia decipiens* or *Sistotrema brinkmanii* had an unconfirmed rot type (Appendix A). 

### 3.2. Xenobiotics Degradation as an Indicator for Fungal Species Remediation Capacity

We developed a rapid screening assay using azo dye DR5B as the model compound to examine the biotransformation capacity of fungal isolates on xenobiotic compounds. It is noted that under the current screening conditions (i.e., 30 °C on PDA plate), the fungal growth rate does not necessarily correlate with the decolorization rate. For example, the fungus may grow slowly yet decolorize the covered area completely. As such, the mycelial growth was measured on a blank plate without dye. 

For the 320 isolates, we obtained 640 phenotypes on the fungal growth rate and dye-decolorization rate. The average dye-decolorization rating for the 320 isolates was at level II. The dye-decolorization rates over a 10-day period allowed us to cluster the fungal isolates into two distinct groups: 63 isolates belonging to 26 species showed fast decoloring rates over time (high group) while 241 isolates from 70 species were rated low decolorization (low group) (Appendix A and Appendix A). In the high group, 38% of isolates completely degraded the dye within 10 days. This group exhibited a diversity of dye-degradation rates at the species level. In the low group, a majority of the isolates (162 isolates) did not exhibit any dye decolorization within 10 days, suggesting that the dye-decolorization phenotype was uncommon in this group under the tested condition. 

For the purpose of the screening assay, we focused on selecting fungal isolates that are fast growers and aggressive dye decomposers. Twenty-nine fungal isolates were not able to grow on the PDA media under the tested conditions, recording a zero-growth level, and 142 out of 320 isolates across 45 fungal species showed different levels of dye-decolorizing ability within the period of screening (Table 1). Moreover, 24 isolates completely decolored the dye plate in 10 days, with mycelial growth that completely covered the plates. Our results showed a strong dependence between growth and decolorization levels (χ² *p*-value = 1.49 × 10^−9^), indicating that biomass is needed for dye decolorization under the tested conditions (Table 1). Among the 142 dye-decolorizing fungal isolates, 16 isolates are brown-rot fungi (29% of all brown-rot fungi screened) and 126 isolates are white-rot fungi (49% of all white-rot fungi screened). 

In the top-24 highest-ranked isolates, the three biggest groups of fungal species that presented aggressive dye decolorization were *Bjerkandera adusta*, *T. versicolor*, and *C. unicolor,* all members of the order Polyporales (Table 2, the comprehensive decolorization index assignment is in Appendix A). These three species had the highest percentage of isolates that degraded RDB5 completely in 10 days. Apparently, dye-decolorization ability does not only vary between species in a predictable way but also varies within species. For instance, all 10 *T. versicolor* isolates were able to decolorize DR5B in 10 days, but the rate of decolorization varied among the isolates (Table 2). This was also the case for *B. adusta* isolates BA6 (HHB-11846-Sp) and BA3 (FP-135159-Sp), which both decolored DR5B in 10 days; however, BA6 had a faster decolorization rate, which decolored the plate completely on day 8 (Figure 3A). At the same time, we examined the growth rates and the dye-decolorization rates in the top-57 aggressive decomposers and found that most of the isolates showed a correlation between growth and decolorization (Figure 3A,B and Appendix A). 

### 3.3. Fungal Species Remediation Traits Variation 

To examine the variations in the dye-decolorizing phenotype for each species, we compared the dye-decolorization index for each species in the entire fungal collection. Twelve species showed a statistically difference from the entire collection (Figure 4 and Appendix A). Ten species showed a higher decolorization ability than the collection average. Two species, *W. cocos* (14 isolates) and *Hypsizygus ulmarius* (9 isolates), exhibited no decolorization during the 10-day testing period. The 10 species with statistically higher decolorization ability showed a small variation in the dye-degradation trait, with an average of more than 95% of the isolates exhibiting decolorization in the fast decomposers (Appendix A). Nevertheless, *T. versicolor* stands out as a species that outperformed all other isolates with its rapid decolorization rate and has a consistent within-species pattern under the tested conditions. The within-species variation analysis potentially indicates a higher probability of identifying rapid-decolorizing strains in the same species. 

### 3.4. Genome-Wide Gene Family Evolutionary Analysis

The dye-decolorization assays showed phenotypic variations between species and within species in our Agaricomycetes collection. To understand the potential genes that are involved in the different dye-decolorization capabilities, we investigated the genomic diversity of 14 Agaricomycetes species, including 8 species with high dye-decolorization capacities and 6 species with low dye-decolorization capacities (Appendix A). The selection is based on the availability of high-quality whole-genome sequences and the decolorization/growth ranks. A total of 17,482 groups of orthologs (orthogroups) were identified, including 1846 single-copy orthogroups, 6338 species-specific orthogroups, and 4408 orthogroups that appeared in all 14 genomes. Moreover, 11,144 orthogroups were shared in at least two species (Appendix A), and 8624 orthogroups were functionally annotated and we grouped them into 3043 functional gene families (Appendix A). 

Carbohydrate catalytic enzymes (CAZymes) and peroxidases assist in the degradation of dyes by breaking down the complex organic compounds. Specifically, cellulolytic enzymes including glycoside hydrolases (GHs) and carbohydrate esterases (CEs) have been found to be associated with dye decolorization [35,36]. Laccase and peroxidases, including Class II peroxidase (POD) and DyP-type peroxidase (DyP), have been characterized to directly decolor the dye molecules [37,38,39]. Laccase and peroxidase have been studied to oxidize the dye molecules, resulting in the breakdown of specific covalent chemical bonds, which leads to molecule degradation. As a matter of fact, the biodegradation of organic compounds typically involves an enzyme network where individual enzymes work synergistically to break down the organic molecule. We thus investigated the CAZyme and the redox enzyme gene families in the genomes of fast and slow decomposers. Auriculariales is considered the origin of the white-rot phenotype and possesses a higher number of genes that facilitate the adaptation to woody plants [40]. We excluded *Exidia glandulosa*, which is in the clade of Auriculariales, due to its long phylogenetic distance from other species and its extended genome size (Figure 5A). Our analysis showed that most gene families showed no differences in gene copy numbers, as they all belong to the same fungal class—Agaricomycetes. However, four gene families, GH10, CE15, POD, and DyP, were statistically larger in fast decomposers than slow decomposers (Table 3 and Appendix A). This result indicated that fungal decomposers enriched in a specific class of CAZymes and peroxidases may have better performance in dye decolorization. Further investigation is needed to understand the metabolic pathways in dye degradation.

To further identify the evolutionary changes of the gene families across species with different dye-decolorization capabilities, we analyzed the genome-wide gene family expansion and contraction from the 3043 gene families in 8 fast decomposers and 5 slow decomposers. CAFE identified 51 gene families that showed significant changes across species (Appendix A). Our results showed that the fungal species with high dye-decolorization capability were expanded in hydrophobin, haem peroxidase, GMC oxidase, chromo-like domain, peptidase, glycoside hydrolase, and dimeric alpha-beta barrel gene families, indicating that lignocellulose decomposition genes and redox genes strongly correlate with dye decolorization (Figure 5B and Appendix A). These genes are known to be expanded in the Polyporales [41]. Interestingly, *Pleurotus ostreatus* is the only fast decomposer in the Agaricales clade. The expanded genes of *P. ostreatus* included fungal-type protein kinase, glucose-methanol-choline oxidoreductase, heterokaryon incompatibility, and cytochrome P450, which indicated that abiotic and biotic stress-responsive genes were expanded in the genomes of fast decomposers. 

## 4. Discussion

Fungi are involved in the deposition of recalcitrant carbon; thus, they have the natural capacity to degrade POPs under the appropriate environmental conditions. Compared to bacteria, fungi are preferable because they have a higher tolerance to environmental toxins based on their diverse ecosystem-related traits [42]. The bioremediation of organic contaminants by different ligninolytic fungal species has been widely studied [15,43], but the capability of bioremediation within species is hardly investigated. We used a model compound, azo dye, to compare the fungal-decolorization capacity and screen a library of 320 fungal isolates across 74 species. The fungal growth and decolorization rate were used as the index to rank all 320 isolates. After screening, we identified eight species as the top tier for genome-wide gene family analysis. We compared the lignocellulolytic gene families and the significantly enriched genes in the top decomposers. The genomic diversity of Agaricomycetes revealed the potential mechanisms that contribute to POP degradation. 

The adaptation to different environmental conditions creates an opportunity to explore the various degradation capacities within fungal species. This study focuses on developing a rapid ranking method to screen large libraries for fungal isolates. The screening revealed variations in traits among different species and identified specific fungi groups that may degrade xenobiotics most rapidly. Phenotypic heterogeneity is significant in fungal biology and ecology, where the fungal strains can survive heterogeneous environments and mount adaptive responses to establish stable populations [44]. As such, the individual fungal strain within the same species can present different capabilities when adapting to changing environments. *T. versicolor* showed a consistent intraspecies ability to degrade dye across all isolates tested while other species exhibited significant within-species variation. For example, one isolate of *T. sanguinea* had a decolorization index of V, while two other isolates of this species did not decolor the plates at all (Appendix A). Nevertheless, we found that the dye-decolorization potential of the isolates tested was strongly dependent on the fungal growth rate (Table 1), different from the finding by Navarro et al., who reported that fungal growth rate is not correlated with dye decolorization [43]. The fast-growing phenotype is associated with a highly competitive activity in fungi [45]; however, the relationship between this phenotype and the degradation of organic compounds may be governed by a complex network. 

Interestingly, we found a correlation between the rapidly expanded gene families identified in the fast decomposers and the overexpressed genes/proteins from our earlier study [46], where *Irpex lacteus* responded to extremely toxic environmental conditions. Carbon metabolism and reduction-/oxidation-related genes (glycoside hydrolase and other hydrolases, NAD(P)-binding domain superfamily) and environmental stress-response genes (cytochrome P450 superfamily) are abundantly present in all fast decomposers (Figure 5B and Appendix A). The same proteins were found to be overexpressed in *I. lacteus* under toxic conditions [46]. The presence of these genes in both studies suggests that strong fungal competitors survive better in the harsh environment, which may positively correlate with the fungal ability to decompose toxic organic chemicals. 

Furthermore, peroxidase, phenoloxidase, and volatile secondary metabolites of C5 to C16 alkanes that contributed to decomposition were reported to be dominant in fast-growing fungi [45]. The fast dye decomposers are also fast growers in our study, which are desired traits for remediation (Figure 3), suggesting that the production of key enzymes required for the fungal decomposition of environmental pollutants may be associated with fungal competitive ability. We found that the gene family hydrophobin was expanded in *T. versicolor*, *P. Ostreatus*, *Fomes fomentarius*, *B. adusta,* and *C. unicolor* genomes, and they were the top fast dye decomposers in our fungal library (Figure 3 and Table 2). Hydrophobin is a secreted fungal-specific protein, forming a hydrophobic layer that facilitates spore dispersal and fungal growth [47,48]. The expansion of hydrophobin in these fast decomposers suggested a potentially fast growth rate. The fast and competitive hyphal growth may result in a high potential for the decomposition of environmental contaminants.

The wood-decay fungal isolates in this study were collected from 38 different states and three countries, representing one of the largest studies of the bioremediation abilities of the Agaricomycetes available in the literature. All the isolates were collected from live or dead wood (Appendix A), where lignocellulose was the carbon source. In a recent study, a screening of 150 strains of basidiomycetes revealed that the most effective decolorizers of azo dyes were within the taxonomic orders of Agaricales and Polyporales, and they were found to produce high levels of laccase [49]. Consistently, we found that gene families of wood-degrading genes such as fungal ligninase (also annotated as haem peroxidase or class II peroxidase) and glycoside hydrolase, involved in lignocellulose decomposition were expanded in the fast dye decomposers (Table 3 and Figure 5B). As such, Nagy et al. proposed from comparative genomic evidence that the genes encoding lignocellulolytic enzymes and the white-rot phenotype first emerged in Auriculariales and evolved in the later clades of Agaricomycetes [40]. The large number of hydrolytic and lignocellulolytic genes in *Exidia glandulosa* genome suggested an acquisition of wood-degrading capacity in Auriculariales [50], where the duplication of wood-decaying genes is not correlated with organic compound decomposition ability. 

Fungal peptidases are hydrolytic enzymes that degrade intracellular and extracellular proteins for nitrogen sources. They are involved in pathogenic invasion, evasion, and escape from the host immune system in pathogenic fungi [51,52], but are rarely studied in wood-decay fungi. Aspartic peptidase is one of the groups that specifically target phenol-containing residue [53]. In the food industry, aspartic proteases were used to remove the phenol-protein complexes that affected the quality of wine and juice [54,55]. The fast decomposers *T. versicolor*, *B. adusta*, *I. lacteus*, and *T. sanguinea* showed significantly expanded in aspartic peptidase domain family (Figure 5B). The expression level of peptidases was also found upregulated in white-rot fungi when growing them on lignocellulosic substrates [41,56,57]. The expansion of aspartic peptidase in the fast dye decomposers suggested a potential direction for further research into the role of peptidases in dye degradation.

## 5. Indications for Future Fungal Remediation

Our study consistently observed a high trait variation with the same species [43]. Regardless of the within-species variations, our study identified several species that have a consistent dye-decolorization capacity. For example, *Trametes versicolor*, *Bjerkandera adusta*, and *Cerrena unicolor* showed a more consistent pattern than other strains in the screening assay. Several expanded genes were identified from the genome-wide gene family analysis. Besides the commonly recognized wood-decaying enzymes that are important for biomass deconstruction, we found that other genes, including hydrophobins and peptidases, were also expanded in the rapid dye-decomposer group. 

Given the enormous diversity in the fungal kingdom, the selection of ideal fungal species and strains for bioapplication is a challenging task. In summary, we presented a semi-quantitative method to screen more than 300 fungal isolates collected from various US locations. From the screening assay, we identified rapid fungal degraders for remediating organic pollutants. We further analyzed the genomic diversity for the rapid and slow degraders and compared that to the slow degraders. The study efficiently identifies fast fungal degraders and suggests that potential gene families could correlate with the fungal-decolorization capacity. However, further investigations are needed to explore all aspects of fungal bioremediation. Firstly, current testing conditions for fungal cultivation are limited to agar plates at 30 °C, which may not be optimal for all fungi. Future research should investigate a broader range of growth temperatures, as different fungi may perform better under different conditions. Secondly, using only one dye molecule (DR5B) in screening assays introduces bias, as it cannot represent the full range of organic pollutants. Including dye molecules with more diversified structural features can reduce this screening bias. Thirdly, current assay procedures only examine two phenotypes—mycelial extension and dye-decolorization—which may not be sufficient to identify fungal species suitable for broad applications. Including other traits such as hyphal density, enzyme production, or toxicity tolerance can increase the screening’s robustness and identify fungal species with greater potential.

## 6. Conclusions

The wood-decaying basidiomycetes have the potential for the bioremediation of organic pollutants in the environment. We evaluated 320 basidiomycetes isolates across 74 species for their growth rate and organic dye-degradation efficiency. We found that the dye-decolorization capacity varies intra- and inter-species. Further, we performed a genome-wide gene family analysis on the top rapid dye-decolorizing fungi species and identified gene families including genes encoding Class II peroxidase, DyP-type peroxidase, CAZymes, reduction-oxidation enzymes, hydrophobin, and secreted peptidases, which were expanded in the fast-decomposer species. Our study provides new insights into persistent organic pollutant removal by fungal isolates at both phenotypic and genotypic levels. However, further research is needed to explore the functional genetic mechanisms of fungal bioremediation and apply diverse conditions for screening a wider range of pollutants. This work contributes to a better understanding of the fungal potential for bioremediation, which can have practical implications for environmental protection and remediation.

## Figures and Tables

**Figure 1 jof-09-00418-f001:**
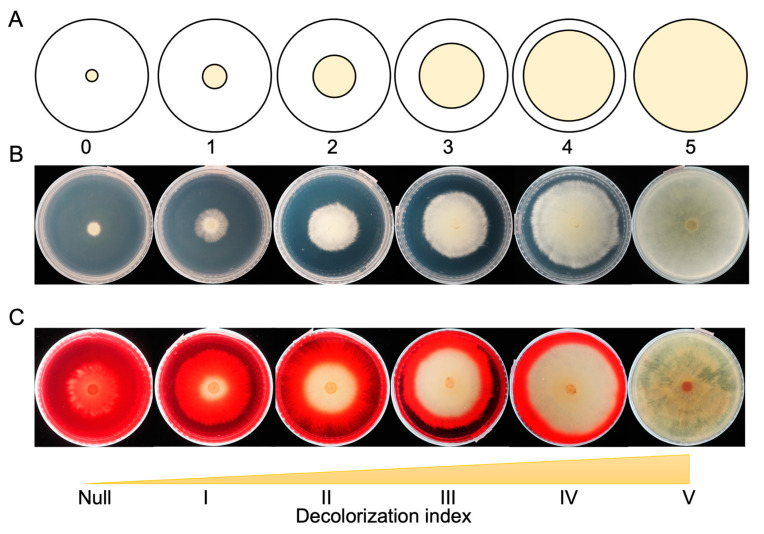
Measurement of fungal growth and dye-decolorization level. (**A**) A diagram of fungal mycelia growth index in the Petri dish. (**B**) Examples of PDA plates with growth indexes from zero to five. (**C**) Examples of PDA+DR5B plates with dye-decolorization indexes from null to V.

**Figure 2 jof-09-00418-f002:**
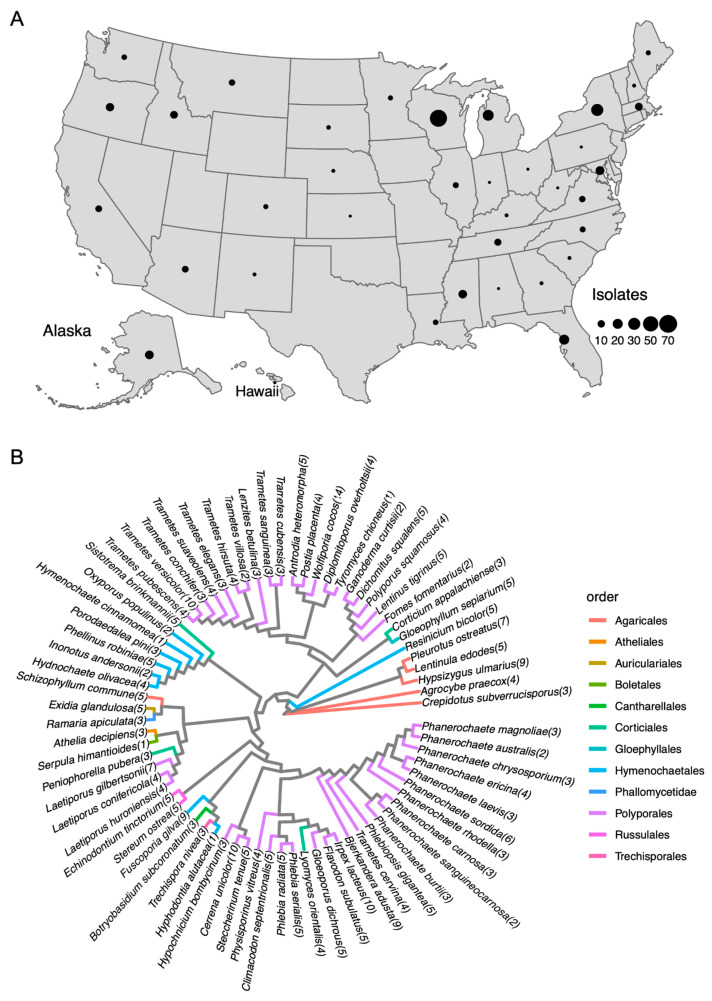
Geographic map and phylogenetic map of the fungal isolate collection. (**A**) 314 out of 320 isolates were collected in the United States across 38 states, including the Virgin Islands, Hawaii, and Alaska. Three isolates from Ontario, Canada, and one from China, are shown on the map. The collection regions of the two isolates were unknown. The sizes of the circles represent the number of isolates collected from the corresponding states. (**B**) Phylogenetic tree of ITS region from the fungal species in this study. The numbers in parentheses following species names indicate the number of isolates in the species. Branches are highlighted with colors corresponding to the order level of classification of each species.

**Figure 3 jof-09-00418-f003:**
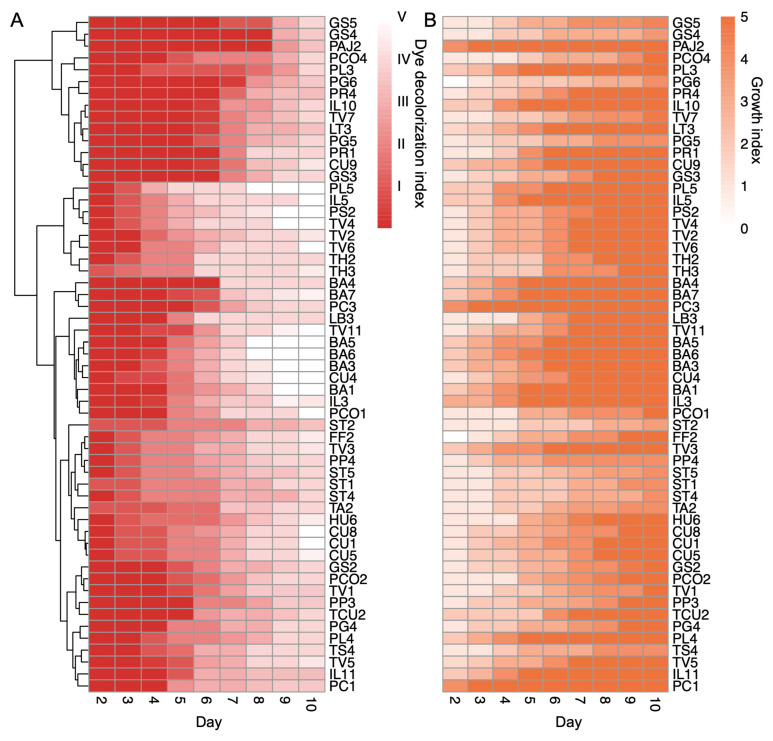
Dye-decolorization levels of the top fungal isolates. (**A**). Heatmap of the top-57 fungal isolates with dye-decolorization levels greater than level III in 10 days. The red color scale indicates the dye-decolorization levels from day 2 to day 10 after inoculation; the fading color suggests the DR5B color is disappearing, with ratings from level I to level V. The fungal isolates were clustered by hierarchical clustering based on the decolorization levels. (**B**). Heatmap of the growth levels in the top-57 dye-decolorizing fungal isolates. The orange color scale indicates growth rates from day 2 to day 10 after inoculation. Both growth and dye-decolorization levels were the means of two biological replicates. The darker color indicates the fungi gain more growth.

**Figure 4 jof-09-00418-f004:**
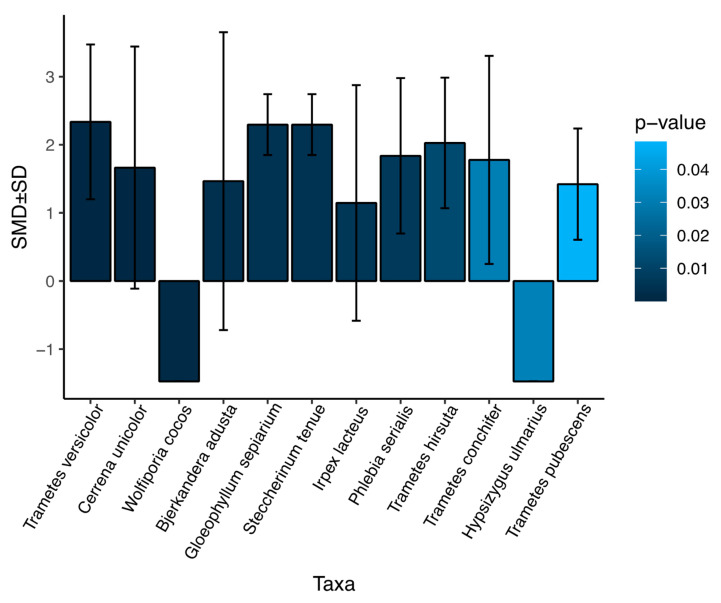
Fungal species with significant variations from the average of all isolates in this study. The difference in dye decolorization for an average of each species compared to the average of all isolates was determined by standardized mean difference (SMD) and *p*-value from the non-parametric Mann–Whitney Wilcoxon test. The variance within species was defined by the standard deviation (SD). The 12 species with a *p*-value < 0.05 were considered statistically different and plotted.

**Figure 5 jof-09-00418-f005:**
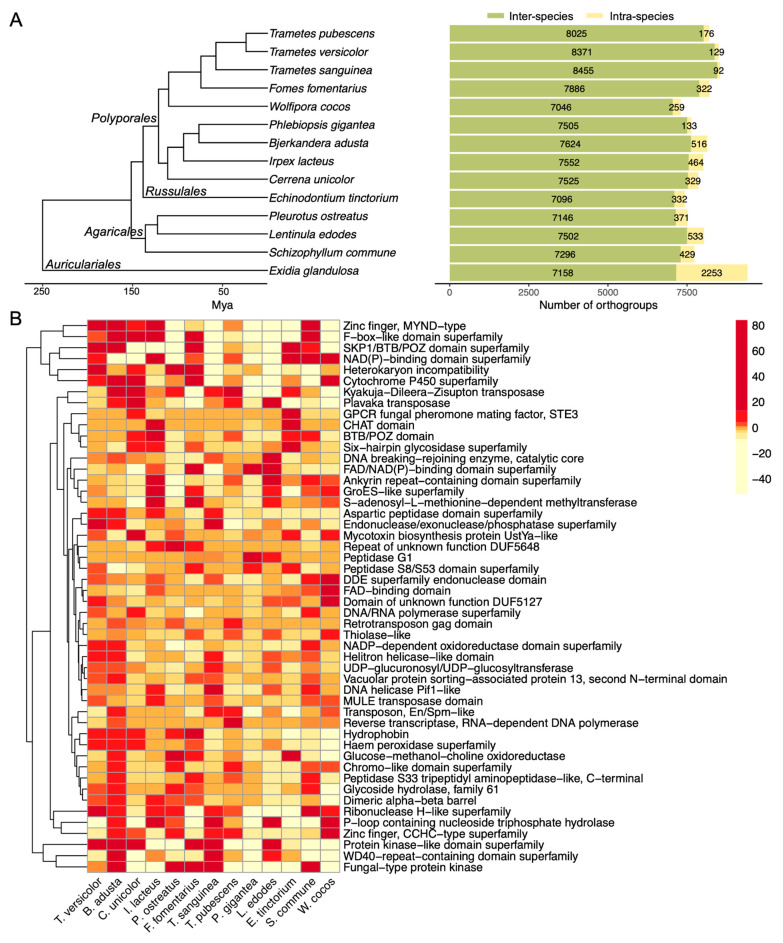
Gene family expansion and contraction in the selected genomes. (**A**) The phylogenetic tree and the number of gene family expansions and contractions in the selected fungal species. The number of inter-species and intra-species orthogroups identified in each genome is shown in the bar plot. Mya, million years ago. (**B**) Significant gene families underwent expansion or contraction. The color scale indicates the changes in gene family expansion or contraction for each species compared to the parent clade. Enzymes identified in the significant gene families are detailed in Appendix A.

**Table 1 jof-09-00418-t001:** Number of fungal isolates, out of 320, assigned a given growth rate or DR5B dye-decolorization level rating after ten days of incubation on PDA or PDA+DR5B, respectively. The numeric value in each cell presents the number of isolates that were assigned with the corresponding decolorization and growth index.

Decolorization Index ------------------ Growth Index	Null	I	II	III	IV	V	Total	Chi-Square Test *p*-Value
0 (No growth)	31	0	0	0	0	0	31	1.49 × 10^−9^
1	35	2	0	0	0	0	37	
2	23	6	4	0	0	0	33	
3	20	1	2	6	0	0	29	
4	20	3	3	4	10	0	40	
5 (Full)	49	16	11	22	28	24	150	
Total	178	28	20	32	38	24	320	

**Table 2 jof-09-00418-t002:** The decolorization levels of fungal species containing the top-24 decomposers. The numbers in the table represent the counts of isolates from selected species corresponding to the decolorization levels observed.

Taxa	*Trametes versicolor*	*Bjerkandera adusta*	*Cerrna unicolor*	*Irpex lacteus*	*Pleurotus ostreatus*	*Phlebia serialis*	*Trametes hirsuta*	*Trametes conchifer*	*Fomes fomentarius*	*Trametes sanguinea*
Ratio of decomposers	10/10	7/9	9/10	9/10	6/7	5/5	4/4	3/3	1/2	1/3 ^a^
Decolorization index										
Null	0	2	1	1	1	0	0	0	1	2
I	0	0	0	2	2	0	0	0	0	0
II	1	1	2	0	2	1	0	1	0	0
III	2	0	1	3	0	1	2	0	0	0
IV	1	1	1	2	1	2	1	1	0	0
V	6	5	5	2	1	1	1	1	1	1

^a^ The ratio of isolates causing decolorization to the total number of isolates in the selected species.

**Table 3 jof-09-00418-t003:** The gene families of wood-decay redox genes in the genomes from the fourteen selected species. Ba (*Bjerkandera adusta)*; Cu (*Cerrena unicolor)*; Ff (*Fomes fomentarius)*; Il (*Irpex lacteus)*; Po (*Pleurotus ostreatus)*; Ts (*Trametes sanguinea)*; Tp (*Trametes pubescens*; Tv (*Trametes versicolor)*; Wc (*Wolfiporia cocos)*; Pg (*Phlebiopsis gigantea)*; Et (*Echinodontium tincorium)*; Sc (*Schizophyllum commune)*; Eg (*Exidia glandulosa)*. Numbers indicate the number of genes in each gene family.

		Fast Decomposers								Slow Decomposers					
Description	Gene Family	Ba	Cu	Ff	Il	Po	Ts	Tp	Tv	Wc	Pg	Et	Le	Sc	Eg
Class II peroxidases	POD	20	19	17	11	9	11	20	26	1	10	4	5	0	37
Dye-decolorizing peroxidases	DyP	14	5	4	5	4	0	3	2	0	5	0	1	0	10
Heme-thiolate peroxidase	HTP	3	1	3	2	3	1	1	1	3	3	4	9	1	32
Multicopper oxidases	MCO	2	17	10	1	11	7	8	10	5	6	15	14	6	12
Copper-radical oxidases	CRO	11	13	23	10	17	9	14	12	4	9	16	7	9	29
Cellobiose dehydrogenase	CDH	2	2	2	1	1	2	2	2	0	2	2	1	1	1

## Data Availability

The ITS sequence for each species are available and deposited to Genbank, with detailed Genbank accessions included in Appendix A. The ITS sequence for each fungal isolate in this study are available on request from the corresponding authors. The data presented in this study are available in Appendix A here.

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
