# Peer review of "Genomic Diversity and Phenotypic Variation in Fungal Decomposers Involved in Bioremediation of Persistent Organic Pollutants"

_jof, 2023, doi:10.3390/jof9040418_

Round 1

Reviewer 1 Report

The present manuscript is very well written and objectives are clear and interesting for JoF and scientists working in the domain.

My major concern is that the paper is based on a genomic study and that there is no functional data as complementary transcriptomic or proteomic data to support the proposed hypothesis. In addition, as it is generally acknowledged, gene annotation is not 100% correct as this is an automatic process will false annotation as data are propagated from genome to genome and not always supported by functional data such as biochemical characterizations. Therefore, I would recommend toning down the different conclusions regarding the assumptions in section 3.4 and the conclusion, i.e. not making a direct conclusion between the genomic data and dye degradation.

- For instance, lines 305-315, the context of this section is very far from the dye decolorization. I'm not sure how useful this section is in this paper.

- lines 317-327, why analyzing the carbohydrate active enzymes in the context of the dye decolorization? There is no link for me. This should be removed.

- in the conclusion, lines 390-393 : “a strong correlation” should be reformulated, explaining that in the fast decomposers, we found abundant…, lines 410-413 and lines 447-450, the same remark as the conclusion is to direct and not confirmed.

Other remarks:

Lines 430- 435 I do not see the link between CB1 and dye decolorization. CB1 are modules that bind to cellulose and are generally grafted to carbohydrate hydrolases. I will remove that part also.

Section 31 : We do not understand why authors have determined the molecular identification of the strain as they are all deposited in a collection. Or this should be explained in the text. Are there corresponding to new collected strains or this is just to confirm their identities?

Minor remarks:

Line 35 : I would not use the term “decomposer” for enzymes which is used for fungi. Please change it.

Lines 39-41 : the definition of brown-rotters is to simple as they do not use only ROS, they have also a partial set of enzymes including hydrolases.

Line 46, I would suggest to start, with “For instance, the decolorization…”.

Line 71, why only lignin peroxidases, and not MnP and VP?

Line 111: please specify as : “ 7 mm dimeter plugs of fungal cultures on PDA…”.

Line 2015-223 : perhaps add a complementary table to cite the top-ranked isolates (this is only a suggestion).

Reviewer 2 Report

This work is significant for development of fungal functions in natural environments. And the authors described a holonomic story about the isolation of functional fungi and dye degradation capacity, I only have four minor comments on this manuscript before it can be accepted for publication.

1. please distinguish enzymes from genes.

2. Can p-value at 0.05 determine the significant difference? P < 0.05 mean a statistical difference but not significance. Especially, as for studies without enough sampling data, or for lab studies with a large heterogeneity in experimental design. p<0.05 is not enough for test the significance between treatments. 

3. Please check the form of reference with three authors cited in text.

4. A conclusion should be added.

Round 2

Reviewer 1 Report

I woul like to thank authors for their corrections that meet the overall remarks. I have no additionnal comments for the paper.